# An automated alarm system for food safety by using electronic invoices

**Wan-Tzu Chang**[1,2]**, Yen-Po Yeh** [3,4]**, Hong-Yi Wu**[2]**, Yu-Fen Lin**[3]**, Thai Son Dinh**[2]**, Ie-bin Lian** [1,2]*

**1** Data Science Research Center, National Changhua University of Education, Changhua, Taiwan, **2** Institute of Statistics and Information Science, National Changhua University of Education, Changhua, Taiwan, **3** Changhua County Public Health Bureau, Changhua, Taiwan, **4** Innovation and Policy Center for Population Health and Sustainable Environment, College of Public Health, National Taiwan University, Taipei, Taiwan

* maiblian@cc.ncue.edu.tw

## Abstract

### Background

Invoices had been used in food product traceability, however, none have addressed the automated alarm system for food safety by utilizing electronic invoice big data. In this paper, we present an alarm system for edible oil manufacture that can prevent a food safety crisis rather than trace problematic sources post-crisis.

### Materials and methods

Using nearly 100 million labeled e-invoices from the 2013–2014 of 595 edible oil manufacturers provided by Ministry of Finance, we applied text-mining, statistical and machine learning techniques to "train" the system for two functions: (1) to sieve edible oil-related e-invoices of manufacturers who may also produce other merchandise and (2) to identify suspicious edible oil manufacture based on irrational transactions from the e-invoices sieved.

### Results

The system was able to (1) accurately sieve the correct invoices with sensitivity >95% and specificity >98% via text classification and (2) identify problematic manufacturers with 100% accuracy via Random Forest machine learning method, as well as with sensitivity >70% and specificity >99% through simple decision-tree method.

### Conclusion

E-invoice has bright future on the application of food safety. It can not only be used for product traceability, but also prevention of adverse events by flag suspicious manufacturers. Compulsory usage of e-invoice for food producing can increase the accuracy of this alarm system.

**Funding:** The work was funded by MOST 106-2314-B-018-001-MY2 from the Ministry of Science and Technology. The funders had no role in study

design, data collection and analysis, decision to publish, or preparation of the manuscript. The e-invoice data was provided by Fiscal Information Agency of Taiwan Ministry of Finance.

**Competing interests:** The authors have declared that no competing interests exist.

## Introduction

The emerging use of rapidly collected, complex data in unprecedented quantities is ushering the world into the era of big data [1]. Although utilization of big data has the potential to afford new insights, improve decision making and governance, and enhance the quality and efficiency of products and services, their application in the food safety domain is still limited [2]. Food safety data and information comprise structured and non-structured data from multiple sectors such as environment, animal, agriculture, food, public health, trade and economy. Previous efforts have explored the predictive power of big data in foodborne illness surveillance, environmental microbial contamination of crops, food safety violations and interpretation of genomic data for tracking and tracing foodborne illnesses [2]. A recent interesting application is that the Chicago Department of Public Health (CDPH) [3] used the data collected from routine food inspections of over 15,000 retail food establishments to train a machine learning model, which produces risk scores for CDPH to prioritize the schedule of inspections.

In past decades, a variety of food fraud incidents have been reported in many countries. Such incidents have had a profound impact on public health and consumer confidence in the safety of food [4]. In response to these incidents, one of the main focuses of food fraud prevention has been on novel prediction models of food fraud using a big data approach, which considered different factors from within and outside the food supply chain. [5–8]. The Bayesian Network by Marvin et al. [6] in 2016 is the first modelling approach or system we can find for the food fraud detection. The approach uses multiple factors on food safety to predict the increased likelihood of occurrence of safety incidents so as to prevent. Bouzembraka et al. [7] developed a food fraud tool called MedISys-FF that collects, processes and presents food fraud reports published globally in the media, which utilize text mining analysis of the articles to facilitate the development of control measures and to detect food fraud. Other approaches like ISAR-Tool (Import Screening for the Anticipation of Food Risks) [8] facilitates a descriptive analysis of the food commodity listed in the national trade statistic, and enables automated detection of unexpected changes in volumes and prices for potential food fraud. Due to the lack of information of business-to-business transactions between manufactures, most of the above are holistic approaches that utilize summarized statistics or reports.

In 2013 and 2014, several food fraud scandals broke out in Taiwan; in particular, contamination and mislabeling of cooking oil were discovered [9], including the use of recycled cooking oil and low quality lard. Although all companies involved were convicted, these incidents caused controversy and severe damage to Taiwan's reputation on food safety.

Despite many detection methods for the identification of adulterated oils and fats have been developed, they have not been adopted as official methods internationally due to their complexity and limited applications [10]. Taiwan Food and Drug Administration had developed analytical methods and "Sanitation Standard for Edible Oils" [11] to assess whether oil has been adulterated. However, the investigation in these incidences detected very few substandard samples. It is fair to say that currently there is little efficient methods on detecting refined adulterated oils from edible oils [12]. Actually the local government had long been suspicious of these problematic manufacturers. Several surprise inspections found that manufacturers were able to manipulate the chemical components of edible oil discreetly so that the composition of fatty acids and other indices matched all the criteria. The convictions on these manufacturers were made not based on the analytic detection of ~~false~~ adulterated chemicals but on their irrational business transaction invoices, i.e., disproportionate amount of materials bought for edible oil to the amount of products sold because they frequently used materials illegal for food production. The result gave us a hint that effective ways to prevent the illegal adulteration of oils may need both on-site inspections and appropriate source management.

The outbreak of major food safety incidence could result costly public panic and damage of goodwill [13]. To address loopholes in existing food safety and traceability rules, the central government of Taiwan passed a law that requires edible oil manufacturers to use electronic invoicing (e-invoice) starting from October 2014 to make transactions of oil merchandise traceable [14].

The Taiwan Ministry of Finance launched an e-invoice system in 2009, and since then, the amount issued increased yearly [15]. In 2018, 7.2 billion business-to-business (B2B) e-invoices were issued and saved in the Fiscal Information Agency (FIA) of the Ministry of Finance. Using invoices in food product traceability systems is not a new idea [16–17]. To establish traceability system, all the related records (ie. invoices, receiving and shipping papers etc.) of each transaction should be kept and retained for a period of five years at least, regardless of whether the form of such records is in paper, electronic or otherwise. To assess the consistency of these records and to identify unusual and inappropriate trends is time-consuming and demanding for experienced manpower because of the heavy burden to inspect the large amount of detailed information concerning date of purchase or supply, name of products, quantity received or supplied, and name and address of the suppliers or distributors etc.

Utilizing e-invoice big data provides an opportunity to overcome aforementioned difficulty. Early detection of food fraud incidents via warning signs of suspicious transactions is a plausible approach. However, based on the authors' knowledge, no literature has addressed automated alarm systems for food safety by utilizing e-invoice big data. An efficient alarm system for edible oil manufacture must be able to (1) sieve the edible oil-related e-invoices of manufacturers who may also produce other merchandise and (2) identify suspicious edible oil manufacture based on irrational transactions from the e-invoice sieved from (1).

Accordingly, this study has two aims: (1) Sieving invoices related to edible oil: The e-invoice big data provided by the FIA can be used to build a classifier based on text mining and machine learning method that can sieve automatically and accurately edible oil-related invoices for each manufacturer and future invoices. (2) Identifying suspicious manufacturers with suspicious monthly transactions: An efficient classifier to identify suspicious manufacturers based on the related e-invoice was shown in **Fig 1**. The above two functions were integrated into an automatic alarm system based on SAS® Enterprise Miner™ 14.3 [18]. In this study, edible oil refers specifically to cooking oil, which is plant, animal, or synthetic fat used in frying, baking, and other types of cooking, as well as in salad dressings.

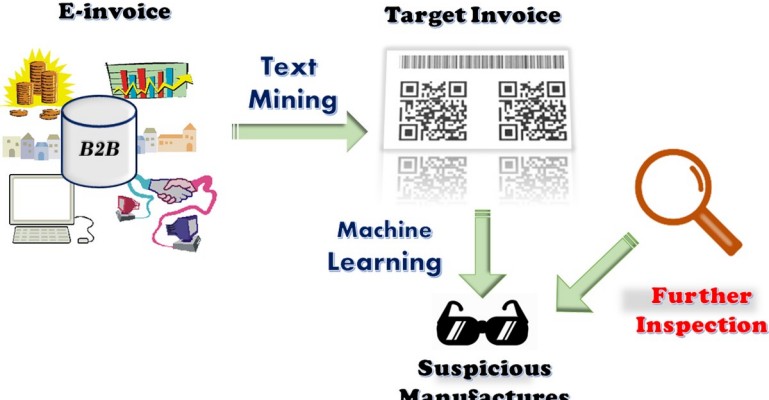

**Fig 1. Tasks of the system: Sieving target invoices and identifying suspicious manufacturers.**

## Methods

### Data processing

A bilateral agreement was signed between FIA and the institute of the authors for permission of accessing the e-invoice data and analyzing it under the supervision of FIA. Data from 99,926,514 e-invoices transacted from January 1, 2014 to October 30, 2017 by 595 edible oil manufacturers were then provided by the FIA. All registered edible oil manufacturers with a capital amount of over 300 million Taiwan dollars (roughly 1 million USD) in Taiwan have been required to use an e-invoice system for all their transactions since 2014 [19] and 595 manufacturers were qualified. Each invoice provided information on the business identities of the vendor and the vendee, including date, amount of money, and item name. However, quantity was not always listed. Given the importance placed on the protection of privacy, the manufacturer names were encrypted. However, according to the list we provided, the FIA sorted them into three categories: A: 21 benchmark manufacturers, B: six problematic manufacturers, and C: 568 unspecified. The A-manufacturers were those commended by the Taiwan Food and Drug Administration before 2014 for their good reputation in high-quality food manufacturing and B-manufacturers were those who had adverse food safety incidence reported or convictions in 2013–2014. The A, B C class each has 10,958,095, 44,590 and 88,923,829 invoices (99,926,514 in total), comprised of 110,448, 10,717 and 970,715 items respectively (over 1 million different items in total). The dataset used in this study is owned by FIA, and was analyzed inside the FIA data center. To access the data researchers can contact Taiwan FIA to apply for the authorization.

### Sieving out edible oil related invoice by text mining

We used the following steps to determine the optimal classifier for sieving out new e-invoices. The flow chart of the steps was illustrated in Fig 2.

**Step 1. Summarizing merchandize name.** The 99,926,514 e-invoices were summarized into 1,011,596 different items in transaction, and among them, 29,942 items were sieved using the keyword "oil". However, they were not yet necessary for edible oil-related items of interest. For example, the Chinese item names of soybean sauce or some cosmetic products also had the keyword "oil" in it, and these should be filtered out. Therefore, we conducted the next steps by sieving further.

**Step 2. Labeling.** We used the eyeballing method to label the 29,942 items into 7,847 "edible oil-related" and 22,095 "non-edible oil-related" products.

**Step 3. Text mining to identify keywords/topics.** The text mining function in the SAS Enterprise Miner automates keywords/topics process using the following steps: ***Text Parsing*** identifies keywords and their instances in data using Nature Language Processing (NLP); ***Text Filter*** eliminates malicious or irrelevant text; and ***Text Topic*** clusters the keywords into m topics, where m is a preset parameter. In this study, we applied m = 60, 90, 120, 150 and 180.

The authors had held several consultation meeting with experts including the Director-general of Changhua County Health Bureau, who has rich experience in dealing with the food safety issues, and managing teams from several manufacturers. We also conducted an expert-knowledge based process to compare with the performance of the above process in which the topics/keywords were selected by artificial intelligence (AI). Researchers selected 60 keywords that they agreed could best distinguish edible oil-related items from others.

**Step 4. Supervised machine learning.** We used the topics or keywords from the above step as features for the following popular supervised learning classifiers: k-nearest neighbor (KNN), support vector machine (SVM), neural network, logistic regression and random forest

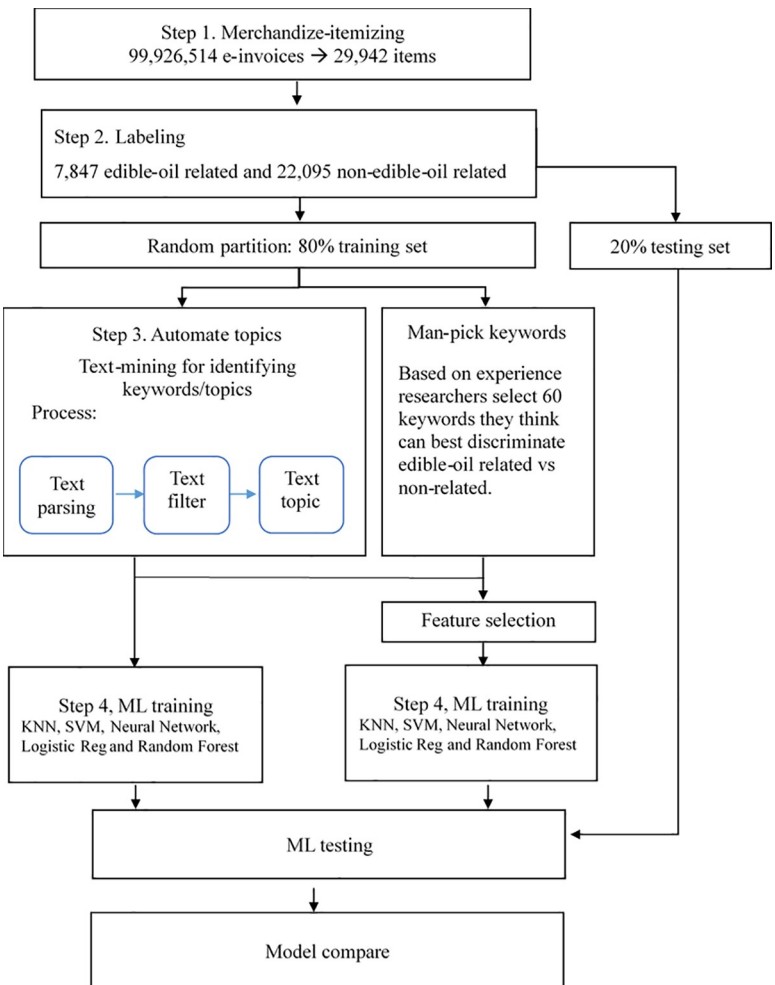

**Fig 2. Flow chart of text mining for sieving out the edible-oil related items.**

(RF). KNN is a decision made by examining the labels on the KNNs and voting [20]. SVMs and neural networks tend to perform better when dealing with multi-dimensions and continuous features [21]. RF is an ensemble classifier that performs well compared to other traditional classifiers for effective classification [22]. We also compared them with a built-in classifier in SAS® EM™ 14.3 called Text Rule Builder. This step was conducted using the with and without feature selection procedure for comparison. The data was randomly divided into training-testing data by 8:2 ratio and results in Tables 1 and 2 were based on the 20% testing data. The classifiers modeled using 5-fold cross-validation strategy. The respective sensitivity and specificity were calculated. The sensitivity refers to the probability of identifying edible oil-related e-invoices. Specificity refers to the probability to identify non-related ones.

**Step 5. Model comparison.** Fig 3 shows the integration of the above steps for sieving related invoices via SAS® EM™ 14.3. The sensitivity and specificity of each classifier were used to determine:

1. Which m is a better selection

2. How the feature selection procedure would improve accuracy

**Table 1. Performance of various classifiers on identifying correct invoices using different choice of m (number of topics).**

| Topic | Accuracy* | KNN (k = 5) | SVM (linear) | Logistic Regression | Neural Network | Random Forest |
|---|---|---|---|---|---|---|
| m:30 | Sensitivity | 91.6% | 81.1% | 81.2% | 88.3% | 93.8% |
| | Specificity | 97.9% | 95.7% | 96.3% | 97% | 98.1% |
| | Error rate | 3.8% | 8.1% | 7.7% | 5.3% | 3% |
| m:60 | Sensitivity | 92% | 84.5% | 85.1% | 91.6% | 95.4% |
| | Specificity | 97.2% | 95.8% | 96.1% | 95.2% | 98.4% |
| | Error rate | 4.1% | 7.2% | 6.8% | 5.1% | 2.4% |
| m:90 | Sensitivity | 91.8% | 87.7% | 89.3% | 91.2% | 95.2% |
| | Specificity | 97.2% | 96.2% | 96.4% | 96.8% | 98.4% |
| | Error rate | 4.2% | 6% | 5.4% | 4.7% | 2.4% |
| | Sensitivity | 92.3% | 90.1% | 91% | 93.5% | 95.2% |
| m:120 | Specificity | 97.5% | 96.2% | 96.9% | 97.1% | 98.5% |
| | Error rate | 3.9% | 5.4% | 4.6% | 3.9% | 2.4% |
| | Sensitivity | 92.4% | 90.4% | 91.4% | 93.2% | 95.2% |
| m:150 | Specificity | 97.3% | 96.2% | 97% | 97.1% | 98.6% |
| | Error rate | 4% | 5.3% | 4.4% | 4% | 2.3% |
| | Sensitivity | 94.1% | 92.1% | 93.1% | 93.6% | 95.2% |
| m:180 | Specificity | 97.9% | 96.7% | 97.6% | 97.9% | 98.5% |
| | Error rate | 3.1% | 4.5% | 3.6% | 3.2% | 2.4% |

*Sensitivity: probability to identify related e-invoice

Specificity: probability to identify the non-related e-invoice

Error rate: total proportion of accuracy

3. Which classifier has the best performance and would apply ensemble techniques on several classifiers to improve performance [23]

## Identifying manufacturers with irrational monthly transaction

Using the edible oil-related e-invoices, we summed up the monthly amounts of purchase and sales for each manufacture. Then, ideally each of the 21 A-labeled and 6 B-labeled manufacturers had records of 46 monthly purchase amount and sales amount. In consultation meetings experts including the Director-general of Changhua County Health Bureau, and managing teams from several A-manufacturers had discussed the proper features on classifying the "rational" vs "irrational" transactions. Based on the longitudinal features of purchase and sales amount and their functions, we applied the supervised learning method including KNN, SVM, neural network, logistic regression, RF, as well as some simple discriminant function build up

**Table 2. Performance of various classifiers on identifying correct invoices using customized keywords.**

| Custom | Accuracy | KNN (k = 5) | SVM (linear) | Logistic Regression | Neural Network | Random Forest |
|---|---|---|---|---|---|---|
| (no feature selection)m:60 | sensitivity | 91.7% | 92.7% | 93.2% | 93.6% | 93.9% |
| | specificity | 97.9% | 98.2% | 98.4% | 98.3% | 98.1% |
| | error rate | 3.8% | 3.2% | 3% | 3% | 3% |
| (feature selection)m:60→31 | sensitivity | 89.7% | 89.8% | 90.2% | 91% | 90.5% |
| | specificity | 97.7% | 97.7% | 97.8% | 97.8% | 97.9% |
| | error rate | 4.4% | 4.4% | 4.2% | 4.4% | 4.1% |

**Fig 3. SAS Enterprise Miner diagram.**

optimal classifiers based on their sensitivity and specificity. Sensitivity refers to the probability of identifying suspicious B manufacturer, while specificity refers to the probability of identifying benchmark A manufacturer. Similar 5-fold and cross-validation strategy in process (1) were applied here.

## Results

### Performance on sieving out correct invoices

Table 1 shows the performance of various classifiers on identifying correct edible oil-related invoices using different choices of $m$ (number of topics). The results were based on 20% of the randomly selected testing data. The error rate stands for the total proportion of accuracy. For SVM, we only listed the results for using linear kernel. For KNN, we only listed k = 5 because they have better performance than other choices of parameters.

The increase of $m$ caused a slightly increase in the sensitivity (se) and specificity (sp) across all methods. Overall, RF has the best performance with se >95% and sp >98%, followed by KNN (se >92% and sp >97%), neural network, logistic regression and SVM. Even the logistic regression had se and sp equal to 91% and 96.9%, respectively when m = 120. The text rule builder can generate an ordered set of rules with se>92% and sp>98%. Certain variations were tested, including adding feature selection steps before classification and several classifiers into one. However, they do not improve the original results, which is already sufficient.

Table 2 lists the results of performance of the same classifiers using 60 keywords selected based on expert knowledge. The results reflected se >93% and sp >98% for most classifiers. Compared with those in Table 1, the machine-automated selection scheme in Table 1 is at least substantial to the expert's knowledge when using more topics (m = 180 for example). This result indicates that while applying this system to other food manufacture monitoring, we can trust the keyword/topic selection ability of the machine without inputting additional expert knowledge. Similar to previous results, reducing keyword numbers (from 60 to 31) by feature selection does not improve the performance.

### Performance on identifying suspicious manufacturers

After summing up the monthly amount of purchase and sales based on the edible oil-related e-invoices of each manufacturer, we further used the series moving averages of the last three months to place the artificial bounds of the month under fuzzification and filter out the noise. In the chain supply, the purchase of a business identity from another identity is referred to as the downstream (D) in contrast to the sales part, which is referred to as the upstream (U). Therefore, we denote the purchase and sales amount of the $i^{th}$ month after smoothing as

follows:

$$D_i = \frac{\text{Pur}_{i-2}}{3} + \frac{\text{Pur}_{i-1}}{3} + \frac{\text{Pur}_i}{3}$$

$$U_i = \frac{\text{Sale}_{i-2}}{3} + \frac{\text{Sale}_{i-1}}{3} + \frac{\text{Sale}_i}{3}$$

We considered that the purchase and sales occurred on the first day of the $i^{th}$ month were not much different from the last day of the previous month. In the results, a total of 425 A-labeled manufacturer month records and 27 B-labeled manufacturer month records are available. A considerable amount of B-labeled records are missing because the compulsory law for using e-invoice started only in 2014 and thus, several B-labeled manufacturers might not have used it before then. After the scandals broke out, certain B-labeled manufacturers were closed, and naturally, the transaction records ceased to exist.

In addition to the original features of $D_i$ and $U_i$, we also tested their various transformation and combination in modeling classifiers to improve its accuracy in classifying "good" and "bad" manufacturers. We found the following two features are the most influential and had relatively better performance in classification:

$$X_1 = log(U_i/D_i) \quad X_2 = log(U_i/D_{i-1}),$$

$X_1$ is the log-transformed turnover ratio of sales to purchase in the same month $i$, and $X_2$ considers the lag effect between purchasing materials and sale products. **Fig 4(A)** shows the scatter plot of A- and B-labeled manufacturer month records, with 95% and 99% elliptic prediction regions, respectively. B-labels appears to cluster on the upper right corner. Apart from KNN, SVM, neural network, logistic regression and RF, we also consider a naïve classifier in the form of a decision tree: *a point is classified as B if $X_1 > 6$ and $X_2 > 6$, and as A, otherwise.*

Table 3 shows the performance of various classifiers on discriminating As and Bs. Evidently, RF had the best performance with se >96% and sp >99%. KNN has the second best performance, follow by the naïve decision tree with simple criteria. These results are useful because the rule can be implemented easily. We also applied the above system to classify the unspecified 1,969 C-labeled manufacturer month records. **Fig 4(B)** presents a scatter plot of adding these unspecified Cs into Fig 4(A). A few Cs were also located at the upper right corner. Using the classifiers trained by As and Bs, roughly 15% of the Cs are classified as suspicious, and these records belong to 13 manufacturers. The results only indicate that 13 manufacturers out of the 569 unspecified C-manufacturers are suspicious enough to be inspected further, and we do not know if they are actually problematic.

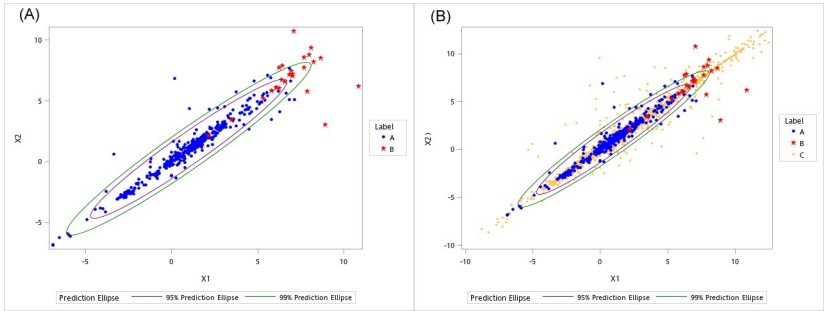

**Fig 4.** (A) Scatter plot of A- and B-labeled manufacturer, (B) Scatter plot of A-, B- and C-labeled manufacturer, with 95% and 99% prediction regions.

**Table 3. Performance on identifying problematic manufacturers.**

|  | Error rate | Sensitivity* | Specificity* |
|---|---|---|---|
| Logistic Regression | 2.65% | 66.67% | 99.29% |
| KNN | 2.21% | 77.78% | 99.06% |
| Neural Network | 2.43% | 66.67% | 99.53% |
| Random Forest | 0.44% | 96.30% | 99.76% |
| SVM (linear) | 5.97% | 0.00% | 100.00% |
| Naïve decision tree | 3.97% | 70.37% | 99.53% |

*Sensitivity: probability to identify B as suspicious manufactures

Specificity: probability to identify A as benchmark manufactures.

## Discussion

To establish traceability system, all the related records (ie. invoices, receiving and shipping papers etc.) of each transaction should be kept and retained for a period of several years, regardless of whether the form of such records is in paper, electronic or otherwise. To assess the consistency of these records and to identify unusual and inappropriate trends is time-consuming and demanding for experienced manpower because it is a heavy burden to inspect the large amount of detailed information concerning date of purchase or supply, name of products, quantity received or supplied, and name and address of the suppliers or distributors etc. Furthermore, since information provided by various suppliers differed with respect to food classifications, product categories and product naming conventions, few regulatory agencies can inspect all the players, upstream and downstream, in the entire food supply chain. Digitalized transactional big data that recorded users' behavior had been applied for human health surveillance, for example, prediction of flu trends [24]. However, no literature had been found on applying electronic invoice for food safety. This study utilizes B2B e-invoice big data to develop an early alert system for edible oil food safety. The system automates the processes of (1) sieving edible oil-related invoices, and (2) identifying suspicious manufacturers with irrational monthly transactions. Processes are based on modeling classifiers via statistical and machine learning methods. In (1), SAS® EM™ 14.3 first automates the construction of m topics from 29,942 merchandise titles of all e-invoices. The assessment on various classifiers shows that based on the 29,942 pre-labeled e-invoices, all classifiers, including KNN, SVM, neural network, logistic regression, RF and text rule builder, can identify accurately the edible oil-related invoice with se >92% and sp >96% with $m \geqq 150$. Particularly, RF has the best performance at se = 95.2% and sp = 98.9% on the testing dataset. The increased performance of a larger number of topics may be because of the diversity of the merchandise name and the tendency of the manufacturer to use catchy names to attract customers.

In (2), RF also has the best performance with se = 96.3% and sp = 99.76%. However, a naïve decision tree with a simple rule to classify whether manufacturer month record is suspicious if $X_1 > 6$ and $X_2 > 6$ also have a reasonably good performance with se = 70.37% and sp = 99.53%. The excess large values of $X_1$, for example, $X_1 > 6$, indicates that the turnover of products sales is $e^6$ (>400) times more than that of material purchase for the specific manufacturer in the same month, while $X_2$ take into account of the lag effect between product sale and materials purchasing from previous month. With both $X_1$ and $X_2 > 6$, the manufacturer had sold products of value 400 times more than the materials it had purchased within a 2-month period. One possible interpretation for this scenario is that a large part of the materials used in products are not considered as proper ingredients for edible-oil manufacture. The increase of se for

naïve decision tree can be achieved by lowering the threshold 6 to smaller value, however as a tradeoff this will reduce its sp resulting extra false alarm.

The drawbacks in this study are as follows:

a. Only 27 B-labeled records are available because of the compulsory law for using e-invoice started only in October 2014. Thus, several B-labeled manufacturers did not use it before then. After the scandals broke out, certain B-labeled manufacturers closed down, and the transaction records became unavailable.

b. The B2B e-invoice data cover only the transactions among domestic companies. The records of materials that manufacturers directly import from overseas were not included in this study. The Taiwanese government is currently working on integrating the Customs records into the FIA system, which will make transactions more complete in the future.

As modern food supply chains become more and more complex, the importance of food production transparency increases. Consumer expectation and right to know has added a challenging dimension to this transparency. Transparency in food supply chains via traceability requires implementation of new technologies such as the Internet of Things (IoT), blockchain and Big Data analytics [25–26]. In response to the food incidences in Taiwan since 2011 [10], Taiwan's Legislative Yuan passed the Act Governing Food Safety and Sanitation in 2013, and enacted new regulations in 2014 [9]. The amendment aim to provide information about food safety as consumers right with efficient trace management of materials, which is based on the registration of their importation, the labeling of additives, the labeling of final products, etc.. Beside that, it also important for government to be able to identify suspicious cases before becoming public event. This alert system can automatically flag suspicious manufacturers for high-priority onsite inspection if implemented into the FIA e-invoice system. Big data analysis can also complement expensive inspections by unveiling numerous unseen issues in traditional methods. However, these promising results should prompt further studies to assess the effectiveness of its application in real-world situations. This study shows that e-invoice has bright future on the application of food safety; not only for product traceability, but also for prevention of adverse events.

## Author Contributions

**Data curation:** Wan-Tzu Chang, Hong-Yi Wu, Thai Son Dinh.

**Methodology:** Wan-Tzu Chang, Yen-Po Yeh, Hong-Yi Wu, Yu-Fen Lin, Ie-bin Lian.

**Project administration:** Ie-bin Lian.

**Software:** Wan-Tzu Chang, Hong-Yi Wu.

**Writing – original draft:** Wan-Tzu Chang, Ie-bin Lian.

**Writing – review & editing:** Ie-bin Lian.

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
