## [Decision Letter · Decision Letter 0]

23 Oct 2019

PONE-D-19-24654

An automated alarm system for food safety by using electronic invoices

PLOS ONE

Dear Prof Lian,

Thank you for submitting your manuscript to PLOS ONE. After careful consideration, we feel that it has merit but does not fully meet PLOS ONE’s publication criteria as it currently stands. Therefore, we invite you to submit a revised version of the manuscript that addresses the points raised during the review process.

The reviewers commented on your manuscript and despite the fact that they considered your work of practical relevance, interesting and potentially novel, there are a couple of shortcomings that should be addressed before it can be acceptable for publication. Please, address the concerns raised by the reviewers.

We would appreciate receiving your revised manuscript by Dec 07 2019 11:59PM. To enhance the reproducibility of your results, we recommend that if applicable you deposit your laboratory protocols in protocols.io, where a protocol can be assigned its own identifier (DOI) such that it can be cited independently in the future. For instructions see: http://journals.plos.org/plosone/s/submission-guidelines#loc-laboratory-protocols

We look forward to receiving your revised manuscript.

Kind regards,

Anderson de Souza Sant'Ana, PhD

Academic Editor

PLOS ONE

Journal Requirements:

2.  Thank you for stating the following in the Acknowledgments Section of your manuscript: "The work was funded by MOST 106-2314-B-O 18-001-MY2

from the Ministry of Science and Technology. The e-invoice data was provided by

Fiscal Information Agency of Taiwan Ministry of Finance."

Please remove any funding-related text from the manuscript and let us know how you would like to update your Funding Statement. Currently, your Funding Statement reads as follows:  The author(s) received no specific funding for this work.

3.  Please could you provide further details about the expert-knowledge based process in section 2.2., step 3:

*If the 'researchers' involved were not the authors, please provide details of their positions and expertise and how they were recruited for this exercise.

*Please clarify if you obtained consent from the researchers to participate in this study.

*Please provide details of how the process was carried out: What questions were the researchers asked? How did they come to an agreement?

Thank you for your attention to these points.

Reviewers' comments:

Reviewer's Responses to Questions

**Comments to the Author**

1. Is the manuscript technically sound, and do the data support the conclusions?

Reviewer #1: Partly

Reviewer #2: Yes

2. Has the statistical analysis been performed appropriately and rigorously? 

Reviewer #1: I Don't Know

Reviewer #2: Yes

3. Have the authors made all data underlying the findings in their manuscript fully available?

Reviewer #1: Yes

Reviewer #2: No

4. Is the manuscript presented in an intelligible fashion and written in standard English?

Reviewer #1: Yes

Reviewer #2: Yes

5. Review Comments to the Author

Reviewer #1: The paper presents a system to detect edible oil manufactures that are suspicious to sell adulterated oils. The system makes use of data mining techniques to detect anomalous individuals in a big sample.

The aim of the paper is interesting and potentially new. Nevertheless, I have seen several shortcomings in the paper, described below. I recommend the authors considering the comments in order to improve the quality of the paper:

1) The contribution is not contextualized. I miss a section dealing with related work about, at least, systems to detect anomalous individuals using data mining techniques. There are many recent contributions in this aspect, for example in financial fraud detection, which is related to this paper. By doing a literature review, the authors can put it clear what the paper adds to knowledge and practice in this line of research.

2) In the Data section, I miss an explanation of how expert knowledge was extracted. Do you interview some key agents? How many? Which was the criterium to select the experts?

3) The selection of the training data should be also discussed. Why not selecting data from a certain time period (e.g., one year) as training data and the following period as testing data? By doing so, the results would show the system ability to predict suspicious manufactures in the future.

4) Which are the specific features influencing the most on the classification? It would be interesting to know and interpret these features.

5) Related to the previous comment, which would be the interpretation of the naïve decision tree in section 3.2? Do you have any explanation about why this rule identifies suspicious manufacturers?

6) Part of the stuff in the Discussion section (that dealing with figure 4(b)) would be better located in the Results section. I also miss the last column in Table 4, commented in the Discussion section.

7) I also recommend including a Conclusion section.

Reviewer #2: I think the manuscript is very interesting and may be very useful to the government from Taiwan. In addition, the method may be inspiring for other countries.

I have some suggestions, as follows:

“Prevention is always better than remedy”- I think the sentence is very colloquial and should be replaced.

“Early detection of food safety events via warning signs of suspicious transactions is thus needed.” – Please give more details about it. Does the e-invoice was already used to detect food adulteration? What are the possible clues to detect a food safety adulteration using the e-invoice system?

“All registered edible oil manufacturers with a capital amount of over 1 million USD in Taiwan have been required to use an e-invoice system” – Can you include the reference about this requirement?

What is FIC? Please include the meaning of abbreviations on their first use.

I think the methods and results sections could be more detailed. The method could be improved to increase the interest from food safety/ food policy specialists and stakeholders

I don’t think Table 1 is necessary. You could include the data in the text.

“Researchers selected 60 keywords that they agreed could best distinguish edible oil-related items from others.” Can you give some examples?

The discussion should be improved. Topics about food safety and consumer’s right could be included. Are there any other studies about using an automated system to detect frauds? The system implementation is expansive?

“Prevention is always better than cure” – same as the sentence aforementioned

Only one reference is from 2017 and one from 2016. The other references are from <2015. I think more recent studies can be included. More references about food issues could be included.

6. PLOS authors have the option to publish the peer review history of their article (what does this mean?). If published, this will include your full peer review and any attached files.

Reviewer #1: Yes: Juan M Hernandez

Reviewer #2: Yes: Diogo Thimoteo da Cunha

---

## [Author Response · Author response to Decision Letter 0]

18 Nov 2019

Journal Requirements:

Reply: 

We revised the manuscript according to PLOS ONE's requirements.

2. Thank you for stating the following in the Acknowledgments Section of your manuscript: "The work was funded by MOST 106-2314-B-O 18-001-MY2

from the Ministry of Science and Technology. The e-invoice data was provided by

Fiscal Information Agency of Taiwan Ministry of Finance."

Please remove any funding-related text from the manuscript and let us know how you would like to update your Funding Statement. Currently, your Funding Statement reads as follows: The author(s) received no specific funding for this work.

Reply:

We remove funding-related text from the manuscript and update the Funding Statement.

“The work was funded by MOST 106-2314-B-018-001-MY2 from the Ministry of Science and Technology. The funders had no role in study design, data collection and analysis, decision to publish, or preparation of the manuscript. The e-invoice data was provided by Fiscal Information Agency of Taiwan Ministry of Finance. We also like to thank Changhua County Public Health Bureau for the constructive discussion.” 

3. Please could you provide further details about the expert-knowledge based process in section 2.2., step 3:

*If the 'researchers' involved were not the authors, please provide details of their positions and expertise and how they were recruited for this exercise.

*Please clarify if you obtained consent from the researchers to participate in this study.

*Please provide details of how the process was carried out: What questions were the researchers asked? How did they come to an agreement?

Reply:

1. All the researchers who involved in data analysis and paper writing were authors.

2. Process: (I) The question “how to efficiently alert food safety issues by using e-invoice” was raised by one of the authors (Dr Yeh), who has served as Director-general of Changhua County Health Bureau for years, and has rich experience in dealing with the food safety issues with related manufacturers. (II) The authors had also hold two meetings with some manufacturer managing teams to discuss about the proper on identifying irrational transaction. (III) the correspondent author (Dr Lian) had signed an agreement with Fiscal Information Agency (FIA) of the Ministry of Finance to access the e-invoice data, and analyze it under the supervision of FIA. 

On 'Data processing' section, we added:

“A bilateral agreement was signed by FIA and the institute of the authors (National Changhua University of Education) for permission of accessing the e-invoice data and analyzing it under the supervision of FIA. “ 

“The dataset used in this study is owned by FIA, and was analized inside the FIA data center. To access the data researchers can contact Taiwan FIA to apply for the authorization.”

On page 11 we added:

“The authors had held several consultation meeting with experts including the Director-general of Changhua County Health Bureau, who has rich experience in dealing with the food safety issues, and managing teams from several A-manufacturers, to discuss the proper features on classifying the "rational" vs "irrational" transactions.” 

 

Reviewer #1: The paper presents a system to detect edible oil manufactures that are suspicious to sell adulterated oils. The system makes use of data mining techniques to detect anomalous individuals in a big sample.

The aim of the paper is interesting and potentially new. Nevertheless, I have seen several shortcomings in the paper, described below. I recommend the authors considering the comments in order to improve the quality of the paper:

1) The contribution is not contextualized. I miss a section dealing with related work about, at least, systems to detect anomalous individuals using data mining techniques. There are many recent contributions in this aspect, for example in financial fraud detection, which is related to this paper. By doing a literature review, the authors can put it clear what the paper adds to knowledge and practice in this line of research.

Reply:

Thank you for your suggestions.

We have added some paragraphs at the introduction section to describe the background of this study regarding food safety and big data. We have also dealt with related works about prediction models of food fraud using big data approach. 

So it reads: “The emerging use of rapidly collected, complex data in unprecedented quantities is ushering the world into the era of big data [1]. Although utilization of big data has the potential to afford new insights, improve decision making and governance, and enhance the quality and efficiency of products and services, their application in the food safety domain is still limited [2]. Food safety data and information comprise structured and non-structured data from multiple sectors such as environment, animal, agriculture, food, public health, trade and economy. Previous efforts have explored the predictive power of big data in foodborne illness surveillance, environmental microbial contamination of crops, food safety violations and interpretation of genomic data for tracking and tracing foodborne illnesses [2-3].

In past decades, a variety of food fraud incidents have been reported in many countries. Such incidents have had a profound impact on public health and consumer confidence in the safety of food [4]. In response to these incidents, one of the main focuses of food fraud prevention has been on novel prediction models of food fraud using a big data approach, which considered different factors from within and outside the food supply chain. [5-8].”

2) In the Data section, I miss an explanation of how expert knowledge was extracted. Do you interview some key agents? How many? Which was the criterium to select the experts?

Reply:

One of the authors (Dr Yeh) has served as Director-general of Changhua County Health Bureau for years, and has rich experience in dealing with the food safety issues with related manufacturers. The authors had also hold two meetings with some manufacturer managing teams to discuss about the proper on identifying irrational transaction. 

On page 11 we added

“The authors had held several consultation meeting with experts including the Director-general of Changhua County Health Bureau, who has rich experience in dealing with the food safety issues, and managing teams from several A-manufacturers, to discuss the proper features on classifying the "rational" vs "irrational" transactions.”. 

3) The selection of the training data should be also discussed. Why not selecting data from a certain time period (e.g., one year) as training data and the following period as testing data? By doing so, the results would show the system ability to predict suspicious manufactures in the future.

Reply: 

We used the 5-fold strategy with 8:2 training-testing-ratio. This is now mentioned on page 10.

We cannot divide the data into training and testing sets by year due to the fact that the compulsory law for using e-invoice started only in October 2014. Thus, several B-labeled manufacturers did not use it before then. After the scandals broke out, certain B-labeled manufacturers closed down, and the transaction records became unavailable. 

4) Which are the specific features influencing the most on the classification? It would be interesting to know and interpret these features. 

Reply:

Among the features we had tried, the combinations of X1 and X2 have relatively better performance: 

X1 is the log-transformed ratio of sales to purchase in the same month i, and X2 considers the lag effect between purchasing materials and sale products. We added the following interpretation in Discussion on page 16-17:

“The excess large values of X1 , for example, X1 >6, indicates that the turnover of products sales is e^6 (>400) times more than that of material purchase for the specific manufacturer in the same month, while X2 take into account of the lag effect between product sale and materials purchasing from previous month. With both X1 and X2 >6, the manufacturer had sold products of value 400 times more than the materials it had purchased within a 2-month period. One possible interpretation for this scenario is that a large part of the materials used in products are not considered as proper ingredients for edible-oil manufacture. The increase of se for naïve decision tree can be achieved by lowering the threshold 6 to smaller value, however as a tradeoff this will reduce its sp resulting extra false alarm.” 

5) Related to the previous comment, which would be the interpretation of the naïve decision tree in section 3.2? Do you have any explanation about why this rule identifies suspicious manufacturers?

Reply: 

We had added the explanation as answered in last question.  

6) Part of the stuff in the Discussion section (that dealing with figure 4(b)) would be better located in the Results section. I also miss the last column in Table 4, commented in the Discussion section.

Reply: 

We had move the part that dealing with figure 4(B) to Results section.

We remove the words “the last column in Table 4”. The result already shown in Fig 4.

7) I also recommend including a Conclusion section.

Reply:

We added a Conclusion section in abstract.

Reviewer #2: I think the manuscript is very interesting and may be very useful to the government from Taiwan. In addition, the method may be inspiring for other countries.

I have some suggestions, as follows:

1.“Prevention is always better than remedy”- I think the sentence is very colloquial and should be replaced.

Reply:

The sentence had been replaced by 

“The outbreak of major food safety incidence could result costly public panic and damage of goodwill [13].”

2.“Early detection of food safety events via warning signs of suspicious transactions is thus needed.” – Please give more details about it. Does the e-invoice was already used to detect food adulteration? What are the possible clues to detect a food safety adulteration using the e-invoice system?

Reply: 

We rephrased the sentence and added a new paragraph on page 6 to explain:

“To establish traceability system, all the related records (ie. invoices, receiving and shipping papers etc.) of each transaction should be kept and retained for a period of five years at least, regardless of whether the form of such records is in paper, electronic or otherwise. To assess the consistency of these records and to identify unusual and inappropriate trends is time-consuming and demanding for experienced manpower because of the heavy burden to inspect the large amount of detailed information concerning date of purchase or supply, name of products, quantity received or supplied, and name and address of the suppliers or distributors etc.

Utilizing e-invoice big data provides an opportunity to overcome aforementioned difficulty. Early detection of food fraud incidents via warning signs of suspicious transactions is plausible approach.”

3.“All registered edible oil manufacturers with a capital amount of over 1 million USD in Taiwan have been required to use an e-invoice system” – Can you include the reference about this requirement?

Reply:

The reference was added on page 8 and in reference [19].

19. Ministry of Health and Welfare/Food Drug Association (MOHW/FDA). Taiwan Food and Drug Administration 2015 Annual Report, page 115. https://www.fda.gov.tw/tc/includes/GetFile.ashx?id=f636694230125946085 Accessed 10.26.19. 

4.What is FIC? Please include the meaning of abbreviations on their first use.

Reply:

We corrected “FIC” to “FIA” and it was first time mentioned with full name on page 6.

FIA is the abbreviation of the Financial Intelligence Agency.

5.I think the methods and results sections could be more detailed. The method could be improved to increase the interest from food safety/ food policy specialists and stakeholders

Reply:

We had add the description of process, including consultation meetings with some manufacturers and agreement with FIA of the Ministry of Finance to access the e-invoice data, and analyze it under the supervision of FIA.

New paragraph about food safety/ food policy and consumer’s right were added in page 4-6 in Introduction as well as in Discussion (page15-17) as requested by other comments.

6.I don’t think Table 1 is necessary. You could include the data in the text.

Reply: 

The content of Table 1 is now included in the text (page 8).

7.“Researchers selected 60 keywords that they agreed could best distinguish edible oil-related items from others.” Can you give some examples?

Reply:

These keywords could be used for inclusion criteria like “Canola (kanola) oil”, “grapeseed”, “sunflower (seed)”, and “cold pressing” etc., or as exclusion like “sauce”, “flavor”, “cosmetic”, and “skin”, etc.. 

8.The discussion should be improved. Topics about food safety and consumer’s right could be included. Are there any other studies about using an automated system to detect frauds? The system implementation is expansive?

Reply: 

We added an paragraph in discussion page 15:

 “To establish traceability system, all the related records (ie. invoices, receiving and shipping papers etc.) of each transaction should be kept and retained for a period of several years, regardless of whether the form of such records is in paper, electronic or otherwise. To assess the consistency of these records and to identify unusual and inappropriate trends is time-consuming and demanding for experienced manpower because it is a heavy burden to inspect the large amount of detailed information concerning date of purchase or supply, name of products, quantity received or supplied, and name and address of the suppliers or distributors etc. Furthermore, since information provided by various suppliers differed with respect to food classiﬁcations, product categories and product naming conventions, few regulatory agencies can inspect all the players, upstream and downstream, in the entire food supply chain. Digitalized transactional big data that recorded users’ behavior had been applied for human health surveillance, for example, prediction of flu trends [24]. However, no literature had been found on applying electronic invoice for food safety.”

And on page 17:

 “As modern food supply chains become more and more complex, the importance of food production transparency increases. Consumer expectation and right to know has added a challenging dimension to this transparency. Transparency in food supply chains via traceability requires implementation of new technologies such as the Internet of Things (IoT), blockchain and Big Data analytics [25-26]. In response to the food incidences in Taiwan since 2011 [10], Taiwan's Legislative Yuan passed the Act Governing Food Safety and Sanitation in 2013, and enacted new regulations in 2014 [9]. The amendment aim to provide information about food safety as consumers right with efficient trace management of materials, which is based on the registration of their importation, the labeling of additives, the labeling of final products, etc.. Beside that, it also important for government to be able to identify suspicious cases before becoming public event.”

9.“Prevention is always better than cure” – same as the sentence aforementioned

Reply:

The sentence had been replaced by

“The outbreak of major food safety incidence could result costly public panic and damage of goodwill [13].”

10. Only one reference is from 2017 and one from 2016. The other references are from <2015. I think more recent studies can be included. More references about food issues could be included. 

Reply: 

We now added the following reference.

1.Wyber R, Vaillancourt S, Perry W, et al. Big data in global health: improving health in low- and middle-income countries. Bull World Health Organ. 2015; 93(3): 203–208. 

doi: 10.2471/BLT.14.139022

2.Marvin HJ, Janssen EM, Bouzembrak Y, Hendriksen PJ, Staats M. Big data in food safety: An overview. Critical Reviews in Food Science and Nutrition. 2017;57(11):2286-2295. 

doi: 10.1080/10408398.2016.1257481

3.Kannan V, Shapiro MA, and Bilgic M. Hindsight Analysis of the Chicago Food Inspection Forecasting Model. Presented at the AAAI Fall Symposium Series (FSS) 2019: Artificial Intelligence in Government and Public Sector. Arlington, Virginia, USA.

4.Spink, J. and Moyer, D. C.. Defining the Public Health Threat of Food Fraud. Journal of Food Science.2011; 76(9):R157-63. doi: 10.1111/j.1750-3841.2011.02417.x.

5.Fritsche J. Recent Developments and Digital Perspectives in Food Safety and Authenticity. Journal of Agricultural and Food Chemistry, 2018;66 (29), 7562-7567.doi: 10.1021/acs.jafc.8b00843

6.Marvin HJP, Bouzembrak Y, Janssen EM, van der Fels-Klerx HJ, van Asselt ED, Kleter GA. A holistic approach to food safety risks: Food fraud as an example, Food Research International. 2016; 463-470. doi: 10.1016/j.foodres.2016.08.028, 89

7.Bouzembrak Y, Steen B, Neslo R, Linge J, Mojtahed V, Marvin H.J.P. Development of food fraud media monitoring system based on text mining, Food Control. 2018; 93; 283-296. 

doi: 10.1016/j.foodcont.2018.06.003

8.Verhaelen, K., Bauer, A., Günther, F., Müller, B., Nist, M., Ülker Celik, B., et al. Anticipation of food safety and fraud issues: ISAR - a new screening tool to monitor food prices and commodity flows. Food Control. 2018;94; 93–101. doi: 10.1016/j.foodcont.2018.06.029

9.Richterich, A.Using Transactional Big Data for Epidemiological Surveillance: Google Flu Trends and Ethical Implications of ‘Infodemiology’ . In: Mittelstadt B., Floridi L. (eds) The Ethics of Biomedical Big Data. Law, Governance and Technology Series. 2016;29: 41-72. Springer, Cham

10.Astill J, Dara RA, Campbell M, Farber JM, Fraser E.D.G, Sharif S, et al. Transparency in food supply chains: A review of enabling technology solutions, Trends in Food Science & Technology. 2019; 91:240-247. doi:10.1016/j.tifs.2019.07.024.

11.Messer KD, Costanigro M, Kaiser HM. Labeling Food Processes: The Good, the Bad and the Ugly, Applied Economic Perspectives and Policy. 2017;39(3): 407–427. doi:10.1093/aepp/ppx028

---

## [Decision Letter · Decision Letter 1]

2 Jan 2020

PONE-D-19-24654R1

An automated alarm system for food safety by using electronic invoices

PLOS ONE

Dear Prof Lian,

Thank you for submitting your manuscript to PLOS ONE. After careful consideration, we feel that it has merit but does not fully meet PLOS ONE’s publication criteria as it currently stands. Therefore, we invite you to submit a revised version of the manuscript that addresses the points raised during the review process.

Your revised manuscript has been reassessed by the reviewers and there are a minor revisions to be done before it can be accepted for publication.

We would appreciate receiving your revised manuscript by Feb 16 2020 11:59PM. To enhance the reproducibility of your results, we recommend that if applicable you deposit your laboratory protocols in protocols.io, where a protocol can be assigned its own identifier (DOI) such that it can be cited independently in the future. For instructions see: http://journals.plos.org/plosone/s/submission-guidelines#loc-laboratory-protocols

We look forward to receiving your revised manuscript.

Kind regards,

Anderson de Souza Sant'Ana, PhD

Academic Editor

PLOS ONE

Reviewers' comments:

Reviewer's Responses to Questions

**Comments to the Author**

1. If the authors have adequately addressed your comments raised in a previous round of review and you feel that this manuscript is now acceptable for publication, you may indicate that here to bypass the “Comments to the Author” section, enter your conflict of interest statement in the “Confidential to Editor” section, and submit your "Accept" recommendation.

Reviewer #1: (No Response)

Reviewer #2: All comments have been addressed

2. Is the manuscript technically sound, and do the data support the conclusions?

Reviewer #1: Yes

Reviewer #2: Yes

3. Has the statistical analysis been performed appropriately and rigorously? 

Reviewer #1: Yes

Reviewer #2: Yes

4. Have the authors made all data underlying the findings in their manuscript fully available?

Reviewer #1: No

Reviewer #2: Yes

5. Is the manuscript presented in an intelligible fashion and written in standard English?

Reviewer #1: Yes

Reviewer #2: Yes

6. Review Comments to the Author

Reviewer #1: The authors have followed some of my recommendations in the first review and now the method and results are clearer. Nevertheless, I need further insist in the first point of my previous review, about the contextualization of the contribution. The authors have framed the work in the literature around food fraud prevention using big data (references 5-8, 24-26). That’s fine, but simply mentioning some references in brackets does not help very much. Your study would be more valued if you make a comprehensive (not necessarily very long) review about the state of the art and what this paper adds to this body of research. In other words, I recommend commenting the cited references (perhaps including new ones), maybe in a new section of related work, highlighting the previous findings and what your study adds.

I have also some other minor comments:

1) Is there any error when citing references 16-17 and 18? It seems reference 16 should be reference 18.

2) Details about the number and role of experts consulted are better located in epigraph “Step 3”, page 9.

3) The quality of the figures should be much improved. I cannot read Fig 3 and Fig 4B.

Reviewer #2: The authors presented a enhanced version of the manuscript.

All my comments were addressed.

The manuscript can be accepted in my opinion.

7. PLOS authors have the option to publish the peer review history of their article (what does this mean?). If published, this will include your full peer review and any attached files.

Reviewer #1: No

Reviewer #2: No

---

## [Author Response · Author response to Decision Letter 1]

3 Jan 2020

Dear Sir:

We had revised the paper according to reviewer’s comments item by item..

Iebin Lian

6. Review Comments to the Author

Reviewer #1: The authors have followed some of my recommendations in the first review and now the method and results are clearer. Nevertheless, I need further insist in the first point of my previous review, about the contextualization of the contribution. The authors have framed the work in the literature around food fraud prevention using big data (references 5-8, 24-26). That’s fine, but simply mentioning some references in brackets does not help very much. Your study would be more valued if you make a comprehensive (not necessarily very long) review about the state of the art and what this paper adds to this body of research. In other words, I recommend commenting the cited references (perhaps including new ones), maybe in a new section of related work, highlighting the previous findings and what your study adds.

Reply:

We thanks you for the advises and had added some descriptions for the most recent literatures [3, 6-8] on page 4 and 5. 

I have also some other minor comments:

1) Is there any error when citing references 16-17 and 18? It seems reference 16 should be reference 18.

Reply:

We corrected citing references 16-18.

2) Details about the number and role of experts consulted are better located in epigraph “Step 3”, page 9.

Reply:

Thank you for your suggestions.

We have added some paragraphs at “Step 3” about the number and role of experts consulted.

3) The quality of the figures should be much improved. I cannot read Fig 3 and Fig 4B.

Reply:

The separate files of figures have higher resolution than those converted to pdf.

---

## [Editor Report · Decision Letter 2]

7 Jan 2020

An automated alarm system for food safety by using electronic invoices

PONE-D-19-24654R2

Dear Dr. Lian,

We are pleased to inform you that your manuscript has been judged scientifically suitable for publication and will be formally accepted for publication once it complies with all outstanding technical requirements.

With kind regards,

Anderson de Souza Sant'Ana, PhD

Academic Editor

PLOS ONE
---

## [Editor Report · Acceptance letter]

8 Jan 2020

PONE-D-19-24654R2 

An automated alarm system for food safety by using electronic invoices 

Dear Dr. Lian:

I am pleased to inform you that your manuscript has been deemed suitable for publication in PLOS ONE. Congratulations! Your manuscript is now with our production department. 

With kind regards,

on behalf of

Professor Anderson de Souza Sant'Ana 

Academic Editor

PLOS ONE